# AsymQ: Asymmetric Q-loss to mitigate overestimation bias in off-policy reinforcement learning

## Abstract

It is well-known that off-policy deep reinforcement learning algorithms suffer from overestimation bias in value function approximation. Existing methods to reduce overestimation bias often utilize multiple value function estimators. Consequently, these methods have a larger time and memory consumption. In this work, we propose a new class of policy evaluation algorithms dubbed, **AsymQ**, that use asymmetric loss functions to train the Q-value network. Departing from the symmetric loss functions such as mean squared error (MSE) and Huber loss on the Temporal difference (TD) error, we adopt asymmetric loss functions of the TD-error to impose a higher penalty on overestimation error. We present one such AsymQ loss called **Softmax MSE (SMSE)** that can be implemented with minimal modifications to the standard policy evaluation. Empirically, we show that using SMSE loss helps reduce estimation bias, and subsequently improves policy performance when combined with standard reinforcement learning algorithms. With SMSE, even the Deep Deterministic Policy Gradients (DDPG) algorithm can achieve performance comparable to that of state-of-the-art methods such as the Twin-Delayed DDPG (TD3) and Soft Actor Critic (SAC) on challenging environments in the OpenAI Gym MuJoCo benchmark. We additionally demonstrate that the proposed SMSE loss can also boost the performance of Deep Q learning (DQN) in Atari games with discrete action spaces.

## 1 Introduction

Learning an accurate value function in Deep Reinforcement Learning (DRL) is crucial; the value function of a policy is not only important for policy improvement (Sutton & Barto, 2018) but also useful for many downstream applications such as risk-aware planning (Kochenderfer, 2015) and goal-based reachability planning (Nasiriany et al., 2019). However, most off-policy DRL algorithms are accompanied by estimation bias in policy evaluation and how to remove this estimation bias has been a long-standing challenge. In this work, we revisit the estimation bias problem in off-policy DRL from a new perspective and propose a lightweight modification of the standard policy evaluation algorithm to mitigate estimation bias.

**Value estimation bias in DRL**  Thrun & Schwartz (1993) firstly shows that maximization of a noisy value estimation consistently induces overestimation bias in Q-learning, where the learned value function overestimates the learned policy, i.e., the prediction from the learned value function is higher than the ground truth value of the policy. Several methods have been proposed to reduce estimation bias in policy evaluation and policy improvement. Hasselt (2010); Van Hasselt et al. (2016) propose double Q-learning, which trains two independent estimators to suppress overestimation. In the continuous state-action space setting, Fujimoto et al. (2018) shows the existence of overestimation in popular Deep Deterministic Policy Gradient (DDPG) (Lillicrap et al., 2015) and proposes Twin Delayed Deep Deterministic Policy Gradient (TD3) to alleviate the overestimation issue, which utilizes the minimum value estimation from two critic networks to fit Q value function. However, these approaches that involve multiple value function estimators together with a minimum operator, may succumb to underestimation bias (Ciosek et al., 2019; Pan et al., 2020), where the value prediction of the learned function is lower than the real policy performance. Lyu et al. (2022); Wu et al. (2020); Wei et al. (2022) Kuznetsov et al. (2020); Chen et al. (2021); Liang et al. (2022);

Lee et al. (2021) propose new critic update schemes to reduce the estimation bias. However, all these methods require multiple actors or an ensemble of critics and usually involve other convoluted tricks that often incur additional computation and memory overheads to improve policy performance. In this work, we explore an efficient approach that reduces value estimation bias without incurring extra computational costs.

**Loss function in policy evaluation** The policy evaluation in DRL is typically based on the Bellman update equation (Bellman, 1966) where the discrepancy between the predicted and target values are minimized. The target value is the combination of the immediate reward and subsequent value function prediction on future states. There are two main components in policy evaluation based on Bellman temporal difference (TD) learning, the fitting target that acts as supervised signals to train neural networks and the loss function that serves as an objective metric. Most of the existing work in reducing estimation bias focuses on the first component, and typically try to construct a more robust target value, such as a lagged target network (Mnih et al., 2015) and ensemble multiple value networks (Fujimoto et al., 2018). The choice of the loss function used for value-function fitting somehow receives much less attention. In practice, most RL algorithms choose symmetric mean square error (MSE) or Huber loss (Patterson et al., 2022) as a metric in fitting value functions. Once zero MSE loss is reached for each state, the prediction of the value network can match policy performance exactly.

In this work, we propose a novel approach based on modifying the loss function to control estimation bias in DRL. We discover that the optimization landscape used to train value networks, governed by the loss functions, plays a crucial role in policy evaluation. A proper choice of the value-fitting loss function can effectively control estimation bias. In particular, we show asymmetric functions can be used for policy evaluation to reduce estimation bias and propose a class of algorithms called **AsymQ**. We specifically evaluate one family of AsymQ loss functions called **SoftMax MSE (SMSE)** to learn the Q value function in both continuous and discrete action environments, but we validate the benefit of using other AsymQ loss functions as well. We find asymmetric loss functions can inject inductive bias to the learning process and thereby control estimation bias present in DRL policy evaluation updates. We highlight the intuition behind our approach in Fig 1, where asymmetry can assign different weights for both overestimated and underestimated states and help alleviate bias in policy evaluation. Notably, compared with existing methods (Hasselt, 2010; Fujimoto et al., 2018), our approach needs only one critic and actor and negligible overhead computational and memory cost.

We summarize our contributions as follows: (1) We show asymmetric loss functions can be used for learning the value function, and introduce a simple asymmetric loss for policy evaluation, namely, Softmax MSE (SMSE) parameterized by a temperature parameter, that can be easily combined with existing RL algorithms. (2) We further propose an auto-tuning algorithm for the temperature to reduce the burden of tuning parameters for our proposed approach. (3) We show that SMSE can significantly reduce estimation bias and improve the performance of popular baseline algorithms such as DDPG on MuJuCo environments and DQN on Atari games. To the best of our knowledge, this is the first algorithm that achieves such competitive performance without the need for other tricks such as multiple critic networks and crafted exploration methods that often incur extra computational and memory costs.

## 2 PRELIMINARIES: VALUE-BASED DEEP REINFORCEMENT LEARNING

The interaction of the RL agent with the environment can be formalized as a *Markov Decision Process* (MDP). The MDP is represented as a tuple $(\mathcal{S}, \mathcal{A}, p, r, \gamma)$, where $\mathcal{S}, \mathcal{A}$ represents the set of states and actions respectively, $p$ represents the transition probabilities, $r(\boldsymbol{s}, \boldsymbol{a})$ the reward function, and $\gamma$ represents the discount factor. The goal of an RL agent is to learn a behavior policy $\pi_\phi(\boldsymbol{a}_t|\boldsymbol{s}_t)$ such that the expected return $J(\pi_\phi) = \mathbb{E}[\sum_{t=0}^{\infty} \gamma^t r(\boldsymbol{s}_t, \boldsymbol{a}_t)|\pi_\phi]$ is maximized.

In RL the value function is usually learned based on temporal difference (TD) learning, an update scheme based on the Bellman equation Sutton & Barto (2018); Bellman (1966). The Bellman equation formulates the value of a state-action pair $(\boldsymbol{s}, \boldsymbol{a})$ in terms of the value of subsequent state-action pairs $(\boldsymbol{s}', \boldsymbol{a}')$:

$$Q^\pi(\boldsymbol{s}, \boldsymbol{a}) = r(\boldsymbol{s}, \boldsymbol{a}) + \gamma \mathbb{E}_{\boldsymbol{s}', \boldsymbol{a}'}[Q^\pi(\boldsymbol{s}', \boldsymbol{a}')], \boldsymbol{a}' \sim \pi(\cdot|\boldsymbol{s}'), \tag{1}$$

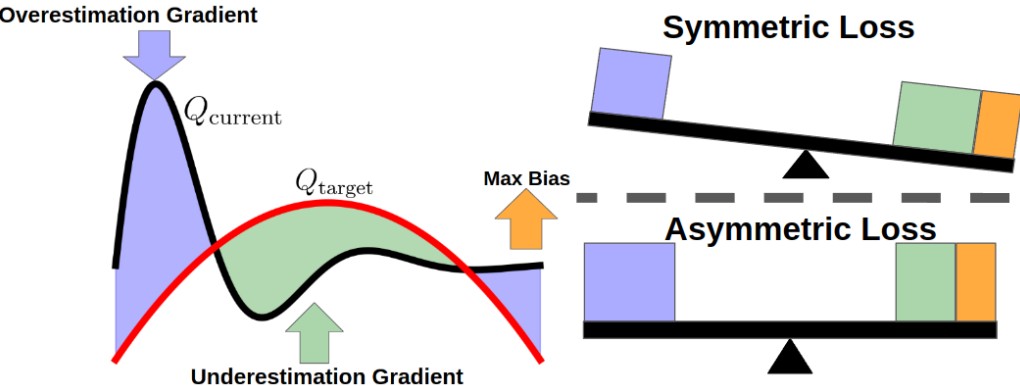

Figure 1: Effect of asymmetric loss on estimation bias in DQN and DDPG. Aside from the gradients from overestimated and underestimated state-action pairs, there exists an additional bias induced by the maximum operator and random noise in policy improvement (Thrun & Schwartz, 1993). Symmetric loss functions assign the same weights for the gradients from overestimated and underestimated state-action pairs thereby leading to overestimation in Q learning and DDPG. Whereas asymmetric loss functions can assign different weights for these even when states-action pairs have the same absolute TD error. The figure on the right illustrates the intuition for why AsymQ loss functions can help counteract the overestimation bias.

where $Q^\pi(s, a) = \mathbb{E}[\sum_{t=0}^{\infty} \gamma^t r(s_t, a_t)|s, a]$, denotes the expected return when performing action $a$ in state $s$ at $t = 0$ and following policy $\pi$ for all the subsequent states. $Q^\pi$ is also known as the value function or the critic.

A policy can be represented and learned explicitly as done in actor-critic methods or inferred implicitly from the value function in methods such as Q-learning. In continuous control, the parameterized policy $\pi_\phi$ can be updated through the deterministic policy gradient algorithm Silver et al. (2014):

$$\nabla_\phi J(\pi_\phi) = \mathbb{E}[\nabla_a Q^\pi(s, a)|_{a=\pi_\phi(s)} \nabla_\phi \pi_\phi(s)]. \tag{2}$$

In DRL, the value function is estimated by a neural network, which we denote as $Q_\theta(s, a)$ parameterized by network parameters $\theta$. Based on temporal difference learning, existing works update $\theta$ by minimizing the mean squared TD-error, $\theta \leftarrow \theta - \alpha \nabla_\theta \mathbb{E}_{s,a}, [\delta(s, a)^2]$ where the TD-error $\delta(s, a)$ is defined by

$$\delta(s, a) = r(s, a) + \gamma Q_{\theta'}(s', a') - Q_\theta(s, a), a' \sim \pi_\phi(s'). \tag{3}$$

Here $Q_{\theta'}$ is usually chosen as a frozen target network. The weights $\theta'$ are usually updated in an exponential moving average approach, i.e., $\theta' \leftarrow \tau\theta + (1 - \tau)\theta'$ for some $0 < \tau < 1$.

We start our investigation by revisiting estimation bias and necessary conditions for fitting the Q value function in Section 3, and show there exist many asymmetric loss functions that also satisfy these conditions. We demonstrate the advantage of using asymmetric loss functions by examining a specific family of AsymQ loss functions that we term as SoftmaxMSE (SMSE) in Section 4 and provide more empirical results and ablation studies in Section 5.

## 3 ESTIMATION BIAS AND ASYMMETRIC LOSS FUNCTION

**Estimation bias and supervised target** The *overestimation* bias in DRL is believed to be due to its greedy update and the noise present in training. The overestimation issue was initially described in Thrun & Schwartz (1993), where the value estimation is updated with a greedy *target* $y = r + \gamma \max_{a'} Q(s', a')$. When there exists noise or approximation error $\epsilon$ in the Q value function, the maximum over it along with the error will generally be greater than the true maximum, namely, $\mathbb{E}_\epsilon[\max_{a'}(Q(s', a') + \epsilon)] \geq \max_{a'} Q(s', a')$. Similar results are also found in the continuous setting. We briefly summarize these results in the following lemma (Fujimoto et al., 2018) (more details are provided in Appendix A).

**Lemma 1.** *For a pair of actor-critic $\{\pi_\phi, Q_\theta\}$, assume one step update based on policy gradient results in a new policy $\pi_{\bar{\phi}}$. Under mild assumptions, the value estimate will be overestimated: $\mathbb{E}[Q_\theta(\boldsymbol{s}, \pi_{\bar{\phi}}(\boldsymbol{s}))] \geq \mathbb{E}[Q^\pi(\boldsymbol{s}, \pi_{\bar{\phi}}(\boldsymbol{s}))]$.*

The overestimation bias turns out to be a serious problem in DRL. The bias may accumulate over iterations and result in an inaccurate value function. Thus, an inaccurate value function will exacerbate sub-optimal actions and lead to poor policies. To alleviate overestimation bias, a popular approach is to construct robust supervised signals from multiple critics to fit the Q function. For instance, TD3 takes a minimum of two critics to update the target in TD-error as

$$\delta(\boldsymbol{s}, \boldsymbol{a}) = y - Q_\theta(\boldsymbol{s}, \boldsymbol{a}), \quad y = r(\boldsymbol{s}, \boldsymbol{a}) + \min\{Q_{\theta_1}(\boldsymbol{s}', \pi_\phi(\boldsymbol{s}')), Q_{\theta_2}(\boldsymbol{s}', \pi_\phi(\boldsymbol{s}'))\}, \quad (4)$$

which can effectively suppress overestimation. However, this pessimistic approach often induces *underestimation* bias (Ciosek et al., 2019; Pan et al., 2020) as we demonstrate in Fig 3. A key challenge in tackling this issue is that we do not know during training if a particular estimate is overestimated as computing the ground truth Q-value is expensive, therefore, we have to rely on proxy TD error signals during training. If the supervised target value is close to ground truth values then overestimated state-action pairs have negative TD errors, while underestimated pairs are associated with positive TD errors.

**Estimation bias and loss function** Another important component of policy evaluation is the loss landscape. MSE of TD errors is the default option for most existing works in learning value functions. Based on Bellman update formula (Bellman, 1966), zero TD-error, i.e., $\delta(\boldsymbol{s}, \boldsymbol{a}) = 0$ for every state-action pair, indicates that the value network can predict value function perfectly. For MSE Q loss, besides its simple formula, the global minima will coincide with a well-fitted value function that has zero TD error. The magnitude of MSE fitting loss quantifies the discrepancy between the learned Q value function and the value prediction target, such as Eq (4). However, the constructed supervised target usually depends on bootstrapping as in Eq (4) and the loss is not the distance from the ground truth value function of the policy. In fact, due to nonconvexity and the limited capacity of neural networks, zero TD error is almost impossible to reach. In practice, finite or even one update step of stochastic gradient descent is employed to optimize value networks after each policy update in DRL.

However, symmetric losses like MSE or Huber are not the only options for optimizing the value network so that zero TD error is reached at its global minimum. In this work, we investigate loss functions that are asymmetric. Compared with MSE loss, in which overestimated states with negative TD error are indistinguishable from underestimated states with positive TD error, asymmetric loss functions can create different loss landscapes for state-action pairs with positive and negative TD errors.

Due to its asymmetry with respect to state-action pairs with positive and negative TD-error $\delta$, we coin our new losses as asymmetric Q (AsymQ) loss functions. Among many choices for AsymQ loss functions, we start our investigation with a simple function that is obtained by making a small modification to the standard MSE function. From MSE $\mathcal{L}_{\text{MSE}}(\{\delta_i\}) = \sum_{i=1}^B \frac{1}{B} \delta_i^2$, we introduce asymmetric loss whose global minimum coincides with zero TD-error

$$\mathcal{L}_{\text{SMSE}}(\{\delta_i\}, T) = \sum_{i=1}^B \mathbf{s.g.}[\text{Softmax}(\{-\frac{\delta_i}{T}\})]\delta_i^2 = \sum_i^B \mathbf{s.g.}[\frac{\exp(-\frac{\delta_i}{T})}{\sum_i^B \exp(-\frac{\delta_i}{T})}]\delta_i^2, \quad (5)$$

where the operator $\mathbf{s.g.}[\cdot]$ denotes the stop gradient operator during backpropagation. The design of this asymmetric loss has several merits. First, it parameterizes a family of loss functions including MSE as a special case when $T \to \infty$. Second, compared with standard MSE loss, it introduces minimum overhead computation and can be interpreted as weighted MSE loss. Instead of assigning the same weights for all samples, Eq (5) determines weights based on the TD-error distribution of samples in one batch. Since the weights in Eq (5) are obtained from Softmax-operator on TD-errors, we coin the asymmetric loss as **Softmax MSE (SMSE)**.

Although Eq (5) depends on the samples in a batch, i.e., a particular sample's weight with the same TD-error can vary across batches, we can still compare samples in batches and their underlying base functions. MSE loss function is based on quadratic function $\mathcal{L}_{\text{MSE}}(\{\delta_i\}) \propto \sum_{i=1}^B \delta_i^2$, while the loss

function for SMSE is (the detailed derivation is included in Appendix A):

$$\mathcal{L}_{\text{SMSE}}(\{\delta_i\}, T) \propto \sum_{i=1}^{B} -\delta_i T \exp\left(-\frac{\delta_i}{T}\right) + T^2\left(1 - \exp\left(-\frac{\delta_i}{T}\right)\right). \qquad (6)$$

This loss landscape is depicted in Fig 2. Note the choice of SMSE is mainly due to its simplicity and versatility as we can control the loss asymmetry and loss landscape by the sign and magnitude of temperature. While in this paper we primarily explore the SMSE loss function, we investigate some additional choices of AsymQ loss functions in Section 5.

Leveraging the asymmetric loss landscape of SMSE, we can effectively encode inductive bias for policy evaluation. For the inherent overestimation bias present in algorithms such as DQN and DDPG, SMSE with positive temperature will penalize states with negative TD error. We illustrate this point with a toy example in Fig 2, where we simulate value function updates with varying SMSE landscapes with all simulations starting with a batch where the TD-errors of all samples are set to a constant negative value by fixing appropriate target values. We study how the TD errors of sampled-batch evolve over updates and find that the loss landscape changes the value function fitting dynamics significantly. Samples in the batch shift from negative TD-error to positive TD-error as we lower the temperature. With the temperature in the right range, estimation bias can be lowered quickly in our synthetic simulation of value function update. We demonstrate that this indeed scales to mitigating estimation bias during RL training on challenging continuous control benchmarks in the next section.

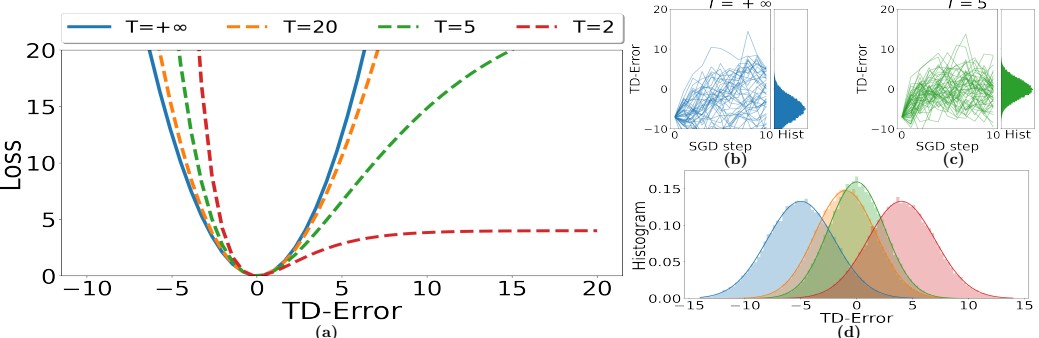

Figure 2: SMSE loss parameterized by temperature $T$ Eq (5) and (6). Though it shares the same global minimum point with MSE, SMSE has a different and asymmetric loss landscape as in Fig 2a. It reduces to MSE when $T = \infty$. Fig 2b,2c simulate SGD trajectories of TD-error on SMSE loss landscape for 10 steps where sampled state-action pairs have a negative TD-error (note that negative TD-error corresponds to overestimation). We further visualize the final histogram of TD-error for different temperatures in Fig 2d. The shaded regions depict histograms for the final TD-error results of each temperature with solid- lines indicating the fitted Gaussian distributions. The intuition found in the toy example can be generalized to more general RL envs, where asymmetric Q loss function can control bias preference of policy evaluation and alleviate inherited overestimation in DDPG and DQN.

## 4   ASYMMETRIC Q LOSS: CONTROL ESTIMATION BIAS

In this section, we show that the small change in value-fitting loss has a large impact on reinforcement learning and that different loss landscapes will lead to different estimation biases. Our simple asymmetric loss can drastically boost policy learning in existing RL methods without involving other computationally expensive tricks. We experiment with our algorithm on the suite of MuJoCo continuous control tasks (Todorov et al., 2012), interfaced through OpenAI Gym (Brockman et al., 2016).

## 4.1 CONTROL ESTIMATION BIAS

We visualize the estimation bias of DDPG, TD3, and DDPG+SMSE (ours) for the OpenAI gym environments Walker2d-v3 and Ant-v3 in Fig 3. The estimation bias is measured by the average difference between Q network prediction and ground truth value for 10,000 randomly sampled action-state pairs. More details regarding how the estimation bias is computed can be found in Appendix B. Clearly, DDPG suffers significant overestimation, which is expected due to inherited maximum bias (see lemma 1). In contrast, TD3 has a serious underestimation issue, a phenomenon that was also reported in Pan et al. (2020); Lyu et al. (2022).

As different temperatures parameterize different loss landscapes, we conduct an ablation study on estimation bias with different temperatures. Lower temperatures put more penalty on action-state pairs whose predictions are larger than their supervised labels and more relaxed for pairs with positive TD-error. As it validates in Fig 3c and 3d, positive temperatures can relieve overestimation issues in DDPG and extremely small positive temperatures even lead to slight underestimation. The results suggest we can to some extent control estimation bias through the parameterized loss function, therefore these asymmetric loss functions can be used to introduce inductive bias for learning value functions. When the temperature is within a proper range, we can significantly reduce estimation bias and learn accurate value functions.

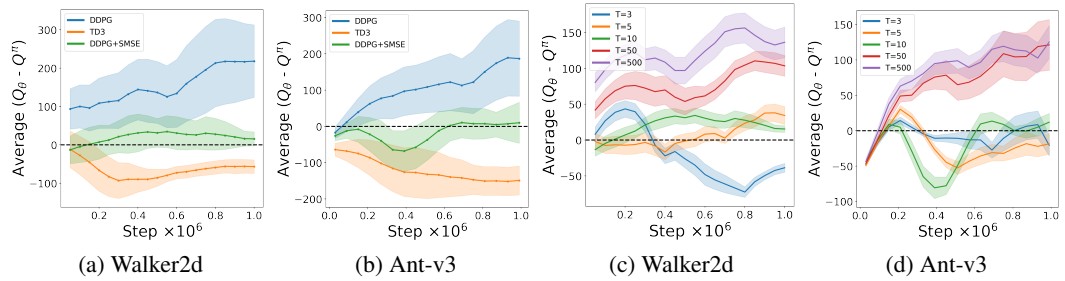

(a) Walker2d      (b) Ant-v3      (c) Walker2d      (d) Ant-v3

Figure 3: Effect of parameterized SMSE loss on estimation bias. Fig 3a and 3b depicts the estimation bias of DDPG, TD3, and DDPG+SMSE in Walker2d-v3 and Ant-v3 respectively. The baseline ($T = \infty$) DDPG with default MSE loss shows a tremendous overestimation effect whereas TD3 exhibits underestimation when compared with their value prediction with ground truth state-action Q value. For SMSE, Fig 3c and 3d illustrates the effect of temperature on estimation bias and highlight the significance of choosing a proper temperature to reduce estimate bias. The shaded region here indicates 0.25 of the standard deviation of the estimation bias.

## 4.2 BOOST POLICY PERFORMANCE

We expect RL algorithms to benefit from more accurate value functions. We investigate the policy performance of DDPG with our SMSE loss. Fig 4a and 4b show the effects of different temperature. As $T$ decreases from positive infinity, the more accurate value function helps boost policy performance. At the same time, we also find learning at extremely small temperatures may deteriorate policy performance. The issue may be explained by the following factors. A lower temperature makes learning more pessimistic as learning may ignore samples with positive TD error and penalizes both true positive and false positive overestimation. As an extreme case, as $T \rightarrow 0$, the weight from softmax in Eq (5) of SMSE will concentrate on samples with the smallest TD-error, which succumb to outliers and noise in the training. To illustrate this effect, we introduce the metric effective batch ratio (EBR),

$$\text{EBR} = \frac{1}{B(\sum_i^B w_i^2)}, \tag{7}$$

where $\{w_i\}$ quantifies the contribution weights for each sample in one batch. Note that for MSE (equivalently SMSE with $T = \infty$) $w_i = \frac{1}{B}$ and EBR $= 1$ so each sample contributes equally to the value function update. On the other extreme, when $T \rightarrow 0$, the weights $\{w_i\} = \text{Softmax}(\{-\frac{\delta_i}{T}\})$ for SMSE concentrate on one sample and EBR $= \frac{1}{B}$. In general, EBR can measure the effective

percentage of samples that contribute to the value function update. EBR is inspired by effective sample size in statistics literature (Murphy, 2012). The statistics also indicate the variance of $\{w_i\}$ in one batch.

We visualize EBR for different temperatures in Fig 4c during training. As expected, EBR decreases as we decrease temperature. We additionally adopt clipped weights in our practical implementation (Algorithm 1) to prevent very small EBR and improve training stability.

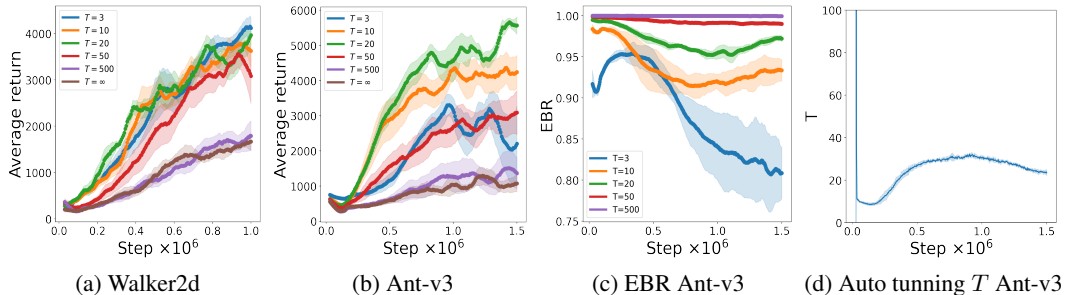

|  |  |  |  |
|---|---|---|---|
| (a) Walker2d | (b) Ant-v3 | (c) EBR Ant-v3 | (d) Auto tunning $T$ Ant-v3 |

Figure 4: Impact of SMSE on policy learning with DDPG. Fig 4a and 4b shows policy performance of DDPG and DDPG+SMSE in Walker2d-v3 and Ant-v3 environments. It depicts that by reducing estimation bias through the parameterized SMSE loss, we are able to boost the learning efficiency and policy performance of DDPG. Fig 4c depicts the effective batch ratio (EBR) for different temperatures of DDPG+SMSE in Ant-v3. Fig 4d illustrates the automatically adjusted temperature over the training steps for DDPG+SMSE* in Ant-v3. The learning curves for DDPG+SMSE* are included in Fig 5 to contrast the performance against fixed temperature SMSE.

### 4.3 AUTOMATIC TEMPERATURE ADJUSTMENT FOR SMSE

The experiments on estimation bias demonstrate the importance of choosing the proper temperature for various environments, and it can be expensive to do an exhaustive grid search for every environment, so auto-tuning of this parameter should greatly reduce the burden of adopting our method. Though there exist many metrics we can employ to perform automatic temperature adjustments (Haarnoja et al., 2018), we present one auto-tuning algorithm tailored for SMSE DDPG with the EBR metric.

In our experiments with DDPG, we find the optimal fixed temperatures for different environments can vary from 3 to 50 for the proposed SMSE loss. An ideal temperature should balance the trade-off between the estimation bias and variance, where a lower temperature puts more penalty on overestimation than underestimation to counteract inherent bias while higher temperatures make the weights uniform which is preferred to resist noise and reduce variance during learning. Here we introduce one simple approach to adjust the temperature based on EBR. Empirically, we find experiments with competitive performance usually have EBR in a fixed interval, which is around $[0.95, 0.98]$. Therefore, a simple tuning scheme can be proposed based on EBR. We increase the temperature when EBR is too small and decrease the hyperparameter when EBR is too large over the expected interval. We summarize the final algorithm in Algorithm 1 (More discussions and details of hyperparameters are included in Section 5 and Appendix B). We term the auto-tuned loss as SMSE*, and illustrate the learning curve and the auto-tuned temperature for DDPG-SMSE* in Ant-v3 in Fig 5 and Fig 4d respectively.

---

**Algorithm 1** AsymQ: Policy evaluation with SMSE-loss with automatic temperature adjustment

---

**Input:** $\beta_{\text{up}}, \beta_{\text{down}}$: interval for desired EBR, $\beta_{\text{multiplier}}$: temperature multiplier

Sample mini-batch of $B$ transitions $(\boldsymbol{s}_i, \boldsymbol{a}_i, r_i, \boldsymbol{s}'_i)$ from replay-buffer $\mathcal{B}$

TD-error $\delta_i = r_i + \gamma Q_{\theta'}(\boldsymbol{s}'_i, \pi_{\phi'}(\boldsymbol{s}'_i)) - Q_\theta(\boldsymbol{s}_i, \boldsymbol{a}_i)$

Weights $\{w_i\} = \text{Softmax}(\text{clip}(\frac{-\{\delta_i\}}{T}, -\ln 2, \ln 2))$

Loss $\mathcal{L}_\theta = \sum_{i=1}^{B} \textbf{s.g.}(w_i)\delta_i^2$ \qquad\qquad\qquad // **s.g.**: stop gradient

Update $\theta \leftarrow \theta - \alpha \nabla_\theta \mathcal{L}$

Every $d$ iterations

    Calculate EBR based on Eq (7)

    $T \leftarrow T \times \beta_{\text{multiplier}}$ if EBR $> \beta_{\text{up}}$

    $T \leftarrow T / \beta_{\text{multiplier}}$ if EBR $< \beta_{\text{down}}$

---

## 5 EXPERIMENTS

We present more experiments to answer the following questions: (1) How is the performance of DDPG+SMSE compared with the existing algorithms with multiple critics? (2) Can our auto-tuning algorithm help find the proper temperature to reduce estimation bias? (3) Can other asymmetric loss functions apart from SMSE improve policy learning? (4) Can our approach work for environments with discrete action space?

### 5.1 MUJOCO ENVIRONMENTS

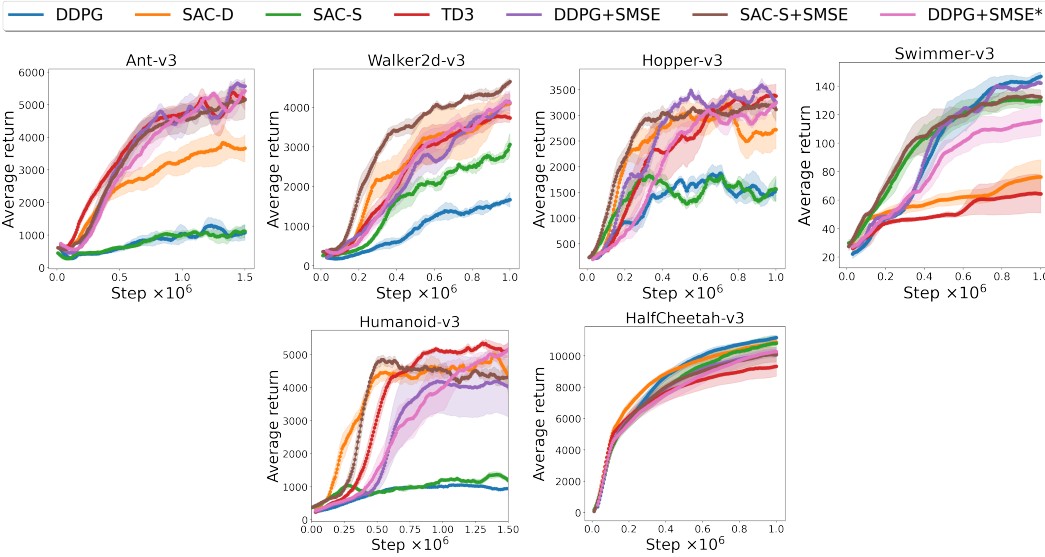

Figure 5: Learning curves for the OpenAI gym continuous control tasks. The shaded region represents half a standard deviation over 5 trials.

**Benchmark** We demonstrate the effectiveness of our approach in the six MuJoCo environments. We compare our approach with popular baselines, DDPG, TD3, and Soft Actor-Critic with automating entropy adjustment (Haarnoja et al., 2018) (SAC). For SAC, we include two variants, SAC-D which uses two critics, and the Clipped Double Q-Learning trick in Eq (4), and SAC-S which only has one critic. For reproducibility, our implementation of selected baselines and our algorithm is based on CleanRL (Huang et al., 2021). We use default hyperparameters unless stated otherwise for all experiments. We apply the SMSE trick in DDPG, denoted as DDPG+SMSE. Our results are

presented in Tab 1 with learning curves in Fig 5. For each environment, we select the temperature by sweeping temperature over values $\{3, 5, 10, 20, 50\}$. We include more details of our implementation and hyperparameters for both our algorithms and compared baselines in Appendix B.

(*RL with a single critic*) The experiment results show our algorithm can significantly improve DDPG by simply replacing MSE loss with the asymmetric SMSE loss for fitting Q functions. Our algorithm achieves performance comparable with the SOTA algorithms with multiple critics. It indicates our algorithm can save half of the computational resources (see Appendix for update time comparisons) for policy evaluation while retaining policy performance, in some cases even outperforming existing algorithms in challenging environments like Ant-v3. (*Automating temperature adjustment*) We also include learning curves obtained by our proposed automating temperature adjustment, denoted as DDPG+SMSE*. Our algorithm can find the proper temperature for most environments. (*Compositionality with other techniques*) We also apply the SMSE trick in SAC-S, denoted as SAC-S+SMSE. The asymmetric loss can also benefit the off-policy algorithm with a stochastic actor. (*Other asymmetric loss function*) Our asymmetric algorithm is not limited to a specific loss. We experiment with two new asymmetric losses:

$$\hat{\mathcal{L}}_1(\{\delta_i\}, T) = \text{Softmax}(\{-\frac{\delta_i}{T}\})\delta_i^2, \quad \hat{\mathcal{L}}_2(\{\delta_i\}, T) = \sum_i T^{-\text{sign}(\delta_i)}\delta_i^2, \ \ (T > 1). \tag{8}$$

Though these new asymmetric losses have different landscapes, they also penalize negative TD-error more and positive TD-error less. Tab 1 shows DDPG with these new asymmetric losses can also boost policy learning of DDPG in challenging environments. Interestingly, we find the fixed temperature $T = 1.5$ for $\hat{\mathcal{L}}_2$ works for all environments.

| Environment | DDPG | SAC-D | SAC-S | TD3 | DDPG+SMSE | SAC-S+SMSE | DDPG+SMSE* | DDPG+$\hat{\mathcal{L}}_1$ | DDPG+$\hat{\mathcal{L}}_2$ |
|---|---|---|---|---|---|---|---|---|---|
| Swimmer-v3 ($\times 10^2$) | **1.47**±**0.06** | 0.76±0.24 | 1.30±0.12 | 0.65±0.27 | 1.43±0.09 | 1.33±0.10 | 1.16±0.21 | 1.21±0.09 | 1.46±0.08 |
| HalfCheetah-v3 ($\times 10^4$) | **1.12**±**0.03** | 1.09±0.04 | 1.08±0.10 | 0.93±0.13 | 1.02±0.05 | 1.01±0.10 | 1.03±0.06 | 1.01±0.11 | 1.07±0.05 |
| Walker2d-v3 ($\times 10^3$) | 1.67±0.37 | 4.10±0.48 | 3.07±0.55 | 3.78±0.33 | 4.15±0.46 | **4.65**±**0.18** | 4.16±0.50 | 3.67±0.25 | 4.26±0.38 |
| Hopper-v3 ($\times 10^3$) | 1.87±0.33 | 3.23±0.26 | 1.83±0.24 | 3.40±0.41 | **3.59**±**0.27** | 3.21±0.16 | 3.24±0.18 | 3.27±0.23 | 3.34±0.31 |
| Ant-v3 ($\times 10^3$) | 1.29±0.49 | 3.83±1.11 | 1.12±0.41 | 5.37±0.48 | **5.66**±**0.25** | 5.15±1.12 | 5.42±0.32 | 5.24±0.44 | 4.98±0.38 |
| Humanoid-v3 ($\times 10^3$) | 1.74±0.11 | 5.00±0.37 | 1.37±0.17 | **5.34**±**0.21** | 5.16±0.20 | 4.85±0.28 | 5.16±0.40 | 4.29±0.43 | 4.95±0.47 |

Table 1: Average Return over 5 seeds with± corresponding to one standard deviation. SMSE* uses automatic temperature adjustment while for SMSE we report best performance from a fixed grid-searched temperature. We also include results for DDPG with asymmetric losses in Eq (8), which validates the benefit of using asymmetric loss functions in general

## 5.2 DQN EXPERIMENTS

We demonstrate that our proposed SMSE loss can also be beneficial for discrete action scenarios by demonstrating it's effectiveness when combined with the DQN algorithmMnih et al. (2015). We evaluate our proposed method on the MinAtar benchmark Young & Tian (2019) and some Atari games Bellemare et al. (2013) comparing it against both baseline DQN and DoubleDQN updateVan Hasselt et al. (2016) with all the variants operating with a common set of hyperparameters. In Fig 6 we report the best-performing learning curves for some of the games in benchmarks obtained by a grid search in (0.5, 1, 5, 10, 20, 100) for the temperature hyperparameter in DQN+SMSE keeping the clip parameter at $\ln 2$. Results for more games in the benchmarks are available in Appendix B.7.

## 6 CONCLUSIONS AND LIMITATIONS

We introduce a lightweight approach to control estimation bias for policy evaluation in DRL. Instead of constructing robust supervised targets at the cost of multiple value networks, we find the fitting loss function in policy evaluation has a surprising effect on controlling estimation bias. We are able to reduce estimation bias and boost policy performance by simply changing MSE loss to an asymmetric loss in DDPG without increasing the computation cost by much. We point out several limitations of our approach here. Though we propose SMSE which includes MSE as a special case and an automatic temperature adjustment algorithm for SMSE, how to design a proper or even optimal loss function is out of the scope of our work. In fact, We actually find SMSE is not so

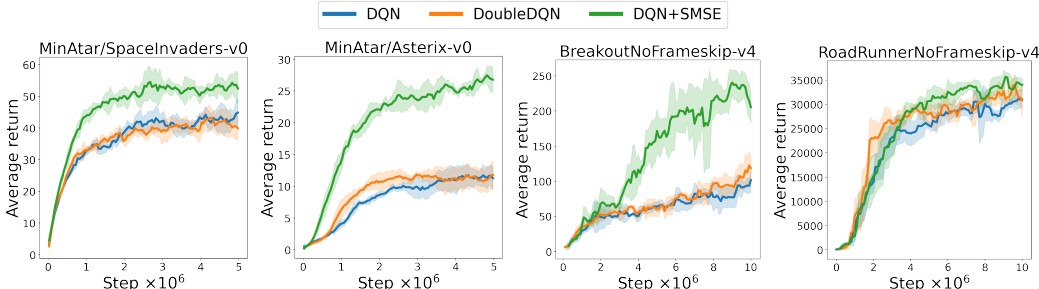

Figure 6: Learning curves for some games in MinAtar and Atari benchmarks. The shaded region represents one standard deviation of the average evaluations over 5 trials.

robust in DQN setting, several fail cases and possible explanations are included in Appendix B. Our results provide one approach to inject inductive bias in DRL via loss functions for policy evaluation. It remains an open problem to adjust and customize the loss landscape automatically for different environments.

## 7 REPRODUCIBILITY STATEMENT

The detailed discussion on assumptions and proofs are included in Appendix A. For experiments, hyperparameters and other implementation details of Algorithm 1 and compared algorithms are included in the main paper and Appendix B. This should provide sufficient information for the reader of interest to reproduce our results.

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

## A PROOF

### A.1 LEMMA 1

The proof is heavily based on conclusions from Section 4.1 and Appendix B from Fujimoto et al. (2018). We summarize the finding below. Given a pair of actor $(\pi_\phi, Q_\theta)$, we can define two new actors from the policy gradient algorithm

$$\bar{\phi} = \phi + \frac{\alpha}{Z_1} \mathbb{E}_{vs}[\nabla_\phi \pi_\phi(s) \nabla_a Q_\theta(s, a)|_{a=\pi_\phi(s)}], \tag{9}$$

$$\hat{\phi} = \phi + \frac{\alpha}{Z_2} \mathbb{E}_{vs}[\nabla_\phi \pi_\phi(s) \nabla_a Q^\pi(s, a)|_{a=\pi_\phi(s)}]. \tag{10}$$

We note $\bar{\phi}$ is based on estimated value function $Q_\theta$ while $\hat{\phi}$ is for ground truth value function $Q^\pi$. $Q^\pi$ is only for analysis purposes and not available during training. We assume $Z_1$ and $Z_2$ are chosen to normalize the gradient. Due to gradient ascend, there exists $\epsilon_1$ such that if $\alpha \leq \epsilon_1$ the estimated value of $\pi_{\bar{\phi}}$ will be bounded below by the estimated value of $\pi_{\hat{\phi}}$:

$$\mathbb{E}[Q_\theta(s, \pi_{\bar{\phi}}(s)] \geq \mathbb{E}[Q_\theta(s, \pi_{\hat{\phi}}(s)] \tag{11}$$

Similarly, there exists $\epsilon_2$ such that if $\alpha \leq \epsilon_2$ then the true value of $\pi_{\bar{\phi}}$ will be bounded above by the true value of $\pi_{\hat{\phi}}$:

$$\mathbb{E}[Q^\pi(s, \pi_{\hat{\phi}}(s)] \geq \mathbb{E}[Q^\pi(s, \pi_{\bar{\phi}}(s)]. \tag{12}$$

If in expectation the $Q_\theta$ is at least as large as the $Q^\pi$ for policy $\pi_{\hat{\phi}}$ $\mathbb{E}[Q_\theta(s, \phi_{\hat{\phi}}(s)] \geq \mathbb{E}[Q^\pi(s, \phi_{\hat{\phi}}(s)]$, then Eq (11) and (12) indicate that policy $\bar{\phi}$ will also be overestimated if step size $\alpha \leq \min(\epsilon_1, \epsilon_2)$:

$$\mathbb{E}[Q_\theta(s, \pi_{\hat{\phi}}(s)] \geq \mathbb{E}[Q^\pi(s, \pi_{\bar{\phi}}(s)]. \tag{13}$$

The above conclusion is possible when we have the normalized gradient during gradient ascent. For the unnormalized gradient, a stronger assumption on $Q^\pi$ and $Q_\theta$ to make Eq (13) hold (Fujimoto et al., 2018). Although Eq (13) is made on one step policy gradient, the overestimation may accumulate as we train policy for more steps.

### A.2 EQ (6)

To prove Eq (6) equivalent to Eq (5) up to normalization constants, we show their gradients with respect to $\delta$ are equivalent.

We denote loss in Eq (5) as $\mathcal{L}_1$ and Eq (6) as $\mathcal{L}_2$. Therefore, the gradients can be obtained

$$\frac{d\mathcal{L}_1}{d\delta_i} = \frac{2 \exp(-\frac{\delta_i}{T})}{\sum_i^B \exp(-\frac{\delta_i}{T})} \delta_i \propto \exp(-\frac{\delta_i}{T}) \delta_i. \tag{14}$$

where $\sum_i^B \exp(-\frac{\delta_i}{T})$ is a normalization constants shared for every sample $\delta_i$ in one batch. On the other hand,

$$\frac{d\mathcal{L}_2}{d\delta_i} = -T \exp\left(-\frac{\delta_i}{T}\right) + \delta_i \exp\left(-\frac{\delta_i}{T}\right) + T \exp\left(-\frac{\delta_i}{T}\right) = \delta_i \exp\left(-\frac{\delta_i}{T}\right). \tag{15}$$

Based on Eq (14) and (15), we can conclude that gradient during one step gradient descent for $\mathcal{L}_1$ and $\mathcal{L}_2$ are equivalent up to normalization constants.

## B EXPERIMENT DETAILS AND DISCUSSIONS

### B.1 IMPLEMENTATION OF SMSE FOR POLICY EVALUATION

For Softmax MSE loss, we include the key modification based on PyTorch (Paszke et al., 2019) in Fig 7. We clip the weight logits for numerical stability. For all experiments, we find clipping range $[-\log 2, \log 2]$ works best for various experiments. $\log 2$ is chosen to guarantee the ratio between the largest weight and the smallest weight is less than 4. Without clipping, our Softmax MSE may assign unreasonable large and small weights for samples in one batch and result in small EBR which leads to very noisy updates and could destabilize the training.

```
td_error = target_q - estimated_q
errors = torch.pow(td_error, 2.0)
# q_loss = errors.mean() ; MSE implementation
+ with torch.no_grad():
+   # clip weights value for numerical stability
+   weight_logits = torch.clip(-td_error / T, math.log(1.0/thres), math.log(thres))
+   weights = torch.softmax(weight_logits) * batch_size
+ q_loss = (weights * errors).mean()
q_loss.backward()
```

Figure 7: PyTorch code sample for modifying MSE loss to SMSE loss with fixed temperature

## B.2 COMPARED BASELINES IN MUJOCO ENVIRONMENTS

For DDPG, SAC-D, and TD3, our implementation is based on CleanRL (Huang et al., 2021) [1], which provides a single-file implementation for each algorithm. We add an extra evaluating step that tests each RL agent in the same environments with 10 different random seeds. All learning curves and reward return numbers are based on experiments in testing environments despite we find that episode return curves during training are similar to testing ones. For DDPG, we use the implementation from Fujimoto et al. (2018) instead of the original DDPG (Lillicrap et al., 2015), which includes several tricks that improve DDPG performance. Our SAC-D implementation matches the performance of the official codebase [2]. For a fair comparison, we use the same learning rate $3 \times 10^{-4}$ and start-timesteps 25000 for all experiments. SAC-S is based on SAC-D but it only uses one critic instead of taking the same minimum value of two independent critics. Unless stated otherwise, the hyperparameters and implementation details follow the default choice in CleanRL.

## B.3 HYPERPARAMETERS FOR SMSE

For SMSE algorithm with fixed temperatures for policy evaluation, we sweep the temperature options across environments. For each environment, we report the performance with the best temperature and present the best performing temperature parameter in Tab 2.

For weights' clip value, we try $(-\ln 2, \ln 2)$, $(-\ln 3, \ln 3)$, $(-\ln 5, \ln 5)$ and $(-\ln 10, \ln 10)$, and find $(-\ln 2, \ln 2)$ and $(-\ln 3, \ln 3)$ can learn good policies most of the time, therefore we use $(-\ln 2, \ln 2)$ by default across environments. In Fig 8, we compare policy performance and EBR for different clipping ranges for a fixed temperature $T = 3$ in Walker2d-V3. The experiment shows that clipping helps stabilize training and learn better policies. We also note that clipping plays a less important role when we use higher temperatures in SMSE.

For auto-tuned temperature SMSE*, we use interval $(0.95, 0.98)$ for desired EBR range in Algorithm 1. The range is selected based on EBR curves for different environments. We have not further swept EBR options and better performance may be possible with fine-tuned hyperparameters.

We believe that the above hyperparameters might not be optimal for SMSE and better hyperparameters can be obtained through a more extensive search. However, our results still demonstrate a surprising performance boost obtained by simply replacing MSE with SMSE, warranting further investigation of asymmetric loss landscapes for value function learning.

**Sensitivity of $\beta_{\text{multiplier}}$** : For Ant-v3 we try three different $\beta_{\text{multiplier}}$ parameters and report the performance curves for DDPG+SMSE* in Fig 9. We see that the performance is robust to the choice of this hyperparameter when contrasted with the difference in performance observed while tuning the temperature parameter directly.

## B.4 OTHER ASYMMETRIC LOSS

We also conduct some initial experiments with more asymmetric loss functions Eq (8). It is worth noting the main difference between $\hat{\mathcal{L}}_1$ and SMSE is that SMSE has a stop gradient operator in calculating SoftMax. Although they appear to be very similar, $\hat{\mathcal{L}}_1$ still presents a different loss landscape and has different gradient descent dynamics. $\hat{\mathcal{L}}_2$ is another even simpler asymmetric

---

[1]https://github.com/vwxyzjn/cleanrl/commit/eba64521299060dd89587a8cee31cba0a8afe930
[2]https://github.com/rail-berkeley/softlearning/pull/127#issuecomment-602748849

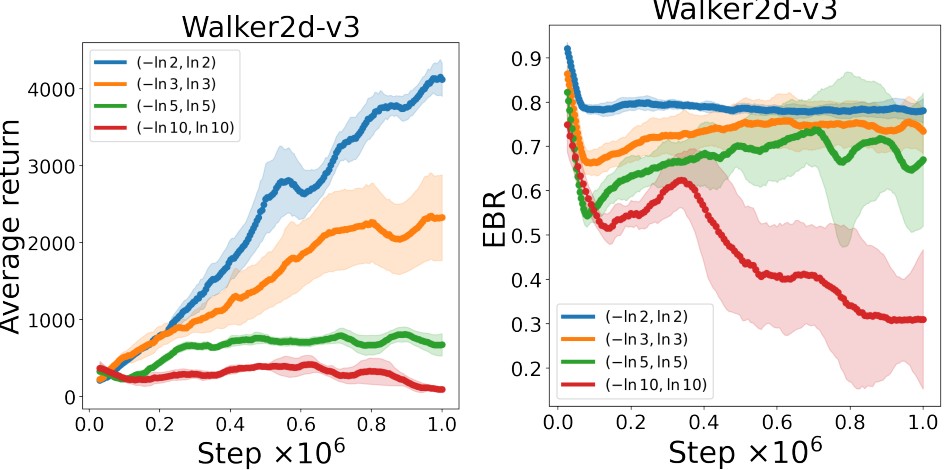

Figure 8: Effect of clipping on policy learning and EBR with $T = 3$. Clipping can help stabilize learning when using SMSE with a lower temperature by avoiding very small EBR. Without clipping, SMSE may assign large weights for overestimated samples and ignore underestimated samples.

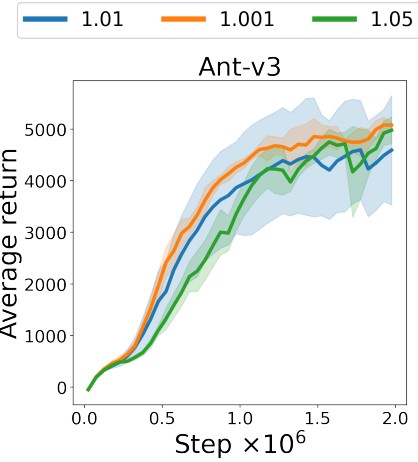

Figure 9: Performance obtained with different $\beta_{\text{multiplier}}$ on Ant-v3 (average over 3 seeds are reported) with DDPG+SMSE* (auto-tune algorithm). We observe reasonable performance for different multiplier hyperparameters demonstrating that the performance sensitivity is lower

| Environment | DDPG+SMSE | SAC-S+SMSE |
|---|---|---|
| Ant-v3 | 20.0 | 5.0 |
| Walker2d-v3 | 3.0 | 5.0 |
| Hopper-v3 | 3.0 | 5.0 |
| Swimmer-v3 | 50.0 | 10.0 |
| Humanoid-v3 | 20.0 | 5.0 |
| HalfCheetah-v3 | 50.0 | 50.0 |

Table 2: Temperature for different environments

loss function, which can be interpreted as a combination of two quadratic functions. We initially parameterize it as

$$\mathcal{L}(\{\delta_i\}, T) = \sum_i T^{-\text{sign}(\delta_i)} \delta_i^2 \qquad (16)$$

We perform a grid-search for $T$ over $\{1.2, 1.5, 2.0\}$, and find that $T = 1.5$ works well for all environments. We emphasize that both functions put more penalization on state-action pairs whose TD-Error is negative.

### B.5 ESTIMATION BIAS FOR FIG 3

For each environment, we pick 20 checkpoints uniformly from the train steps to test RL agents' performance. For each checkpoint, we evaluate the learned Q values on 10,000 state-action pairs that are uniformly sampled from the replay buffer. The predicted Q-value is obtained by simply feed-forwarding the state-action pair into the Q-network from the checkpoint. The ground truth $Q^\pi$ is estimated in Monte-Carlo fashion, where we execute the policy checkpoint and collect rewards along trajectories to estimate the average discounted return $J(\pi)$, the trajectories are clipped at 500 steps and the average over 3 such trajectories are treated as the ground truth value, we use only 3 trajectories because the variance observed across trajectories were low as the policy is deterministic. We plot the mean and standard derivation of estimation bias over 10,000 state-action pairs in Fig 3. The pseudo-code for this procedure is provided in Algorithm 2.

---

**Algorithm 2** checking estimation bias in the value function for a given checkpoint

---

Let $\pi$, $Q^\pi$ be the policy and value function to evaluate.
Explore environment with $\pi + \mathcal{N}(0, \epsilon)$ and store $N$ transitions to replay-buffer $\mathcal{B}$
Sample M state-action pairs $\{(s_i, a_i)\}$ from $\mathcal{B}$
Let Q_PRED = $\begin{bmatrix} 0 \end{bmatrix}_{1 \times M}$ and Q_TRUE = $\begin{bmatrix} 0 \end{bmatrix}_{1 \times M}$
**for** i = 1 **to** M **do**
    Q_PRED[i] = $Q^\pi(s_i, a_i)$
    Q_TRUE[i] = Monte-Carlo-estimate(discounted_return($s_i, a_i, \pi$))
**end for**
**return** mean and standard deviation of $\left( \text{Q\_PRED - Q\_TRUE} \right)$

---

### B.6 UPDATE TIME ANALYSIS

We benchmark the average update time for the algorithms DDPG, TD3, SAC-D, SAC-S, DDPG + SMSE, and SAC-S + SMSE for training on the Ant-v3 environment. All algorithms are run with the same batch-size hyperparameter on an Intel i9-9900 16 core + RTX 2080 Ti desktop. From Fig 10, we observe that our SMSE loss function adds very minimal cost (0.2 ms/step) to the update time on-top of DDPG but still achieves performance comparable to the state-of-the-art algorithms.

### B.7 ENVIRONMENTS WITH DISCRETE ACTIONS

The DQN implementation for the Atari benchmark is based on the Tianshou(Weng et al., 2021) package that leverages envpool(Weng et al., 2022) implementations of the Atari environments to speed up the run-time. The architecture and pre-processing of frames are similar to the ones found in the original DQN paper(Mnih et al., 2015). The hyperparameters used to come from the default parameters present in the Tianshou code-base, we just experimented with different temperature parameters of the additionally introduced SMSE loss, and temperatures from the set (0.5, 1, 5, 10, 20, 100) were tried. In Tab 3 we outline the best performing run's temperature and final performances for Atari games and contrast it against the baseline DQN (T = $\infty$) and DoubleDQN. The results reported are the mean and standard deviation of returns over 5 random seeds.

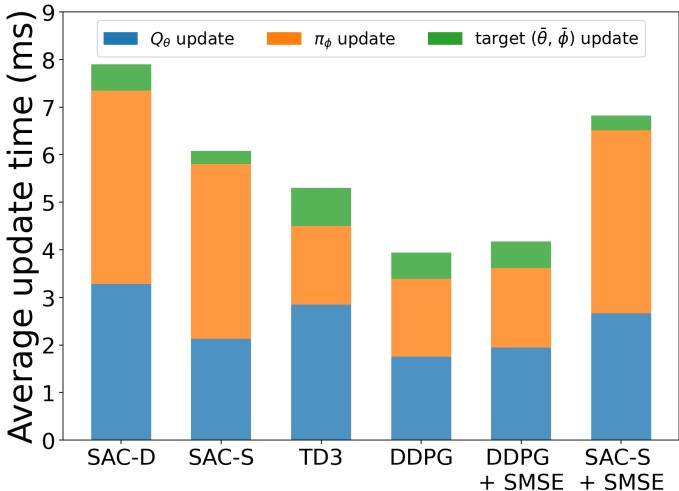

Figure 10: Average update time for each training step for different algorithms. DDPG + SMSE adds very little computational cost over DDPG but achieves performance comparable to that of TD3 or SAC

| Environment | Temperature | DQN + SMSE | DQN | DoubleDQN |
|---|---|---|---|---|
| Breakout | 1 | **205.31±21.52** | 101.56±15.70 | 118.64±17.35 |
| SpaceInvaders | 1 | **596.75±41.89** | 533.14±96.60 | 502.17±75.90 |
| Seaquest | 1 | 2920.42±413.79 | **2939.12±389.32** | 2904.19±354.87 |
| RoadRunner | 10 | **33990.19±1235.37** | 31016.93±1786.06 | 30798.37±3308.80 |
| BeamRider | 20 | **3379.35±611.30** | 3368.27±571.66 | 2026.12±465.88 |
| Riverraid | 100 | 7463.80±1002.75 | 7651.94±681.38 | **8274.78±778.29** |
| Asterix | 10 | **2983.21±685.62** | 2714.63±407.57 | 2079.31±344.00 |

Table 3: Temperature for best-performing DQN + SMSE Loss on Atari games benchmark (Gym:NoFrameskip-v4). We report the average Return over 5 seeds with ± corresponding to one standard deviation.

For the MinAtar (Young & Tian, 2019) games benchmark, we use the CleanRL (Huang et al., 2021) implementation of DQN and leverage the hyperparameters suggested in the MinAtar paper along with additionally limiting the environment steps for the *MinAtar/Seaquest-v0* environment to 10,000 steps as done in (Lan et al., 2020). Similar to the Atari games benchmark we try different temperature parameters from the set (0.5, 1, 5, 10, 20, 100) and report the best performing run in Tab 4, where the result represents the mean and standard deviation over 5 random seeds.

| Environment | Temperature | DQN + SMSE | DQN | DoubleDQN |
|---|---|---|---|---|
| MinAtar/Asterix-v0 | 1 | **26.79±1.85** | 11.26±1.85 | 11.77±2.19 |
| MinAtar/SpaceInvaders-v0 | 1 | **52.48±4.74** | 44.86±4.25 | 39.92±3.84 |
| MinAtar/Breakout-v0 | 20 | **17.53±2.28** | 17.15±0.94 | 14.98±0.84 |
| MinAtar/Seaquest-v0 | 10 | **4.56±0.73** | 4.53±1.41 | 3.24±0.53 |
| MinAtar/Freeway-v0 | 20 | 48.60±0.41 | **48.74±0.52** | 48.07±1.44 |

Table 4: Temperature for best-performing DQN + SMSE Loss on MinAtar benchmark. We report the average Return over 5 seeds with ± corresponding to one standard deviation.

While the performance improvement is observed on some of the games in both the Atari and MinAtar benchmarks, it still fails to show major improvements for a good fraction of the environments, unlike the continuous control MuJoCo benchmark. This could be due to several reasons, one major factor is that estimation bias in DQN is different from that of DDPG, and the loss developed and

| Environment | Algorithm | Performance | Steps | Wall-clock Time (hours) | Estimated Total FLOPs ($\times 10^{18}$) |
|---|---|---|---|---|---|
| Hopper-v3 | REDQ | 3528.57±134.14 | 125K | 20.46 | 3.303 |
| Hopper-v3 | SAC-S+SMSE | 3213.84±205.56 | 1000K | 11.93 | 0.161 |
| Walker2d-v3 | REDQ | 5426.60±626.55 | 300K | 50.32 | 8.291 |
| Walker2d-v3 | SAC-S+SMSE | 4653.17±187.51 | 1000K | 11.21 | 0.168 |
| Ant-v3 | REDQ | 6086.05±122.02 | 300K | 53.31 | 11.144 |
| Ant-v3 | SAC-S+SMSE | 5172.35±112.12 | 1500K | 16.16 | 0.338 |

Table 5: Comparison against REDQ (Chen et al., 2021). Our SAC-S+SMSE can achieve competitive results with much fewer computational resources measured in wall-clock time and estimated FLOPs count.

tested extensively in DDPG doesn't simply transfer and work for all DQN environments. It is to be noted that even DoubleDQN updates sometimes result in lower performance compared to DQN so it can be expected that just reducing estimation bias does not always necessarily translate to higher performance. Additionally, we leverage the hyper-parameters suggested in popular open-source packages and do not perform any tuning apart from a small grid search on the SMSE-temperature parameter. An extensive hyper-parameter search along with the exploration of new asymmetric loss functions for environments with discrete action spaces is an exciting direction for future work.

### B.8 SMSE WITH ENSEMBLE Q-NETWORKS

Arguably one of the most effective approaches to reducing the bias and having stable value-function targets is the usage of an ensemble of critics (Kuznetsov et al., 2020; Chen et al., 2021; Liang et al., 2022; Lee et al., 2021; Lyu et al., 2022; Wu et al., 2020; Wei et al., 2022). In this section, we compare SMSE with state-of-the-art algorithms based on an ensemble of critics. We find our approach can achieve competitive results in a much more computational efficiency approach. Besides, we surprisingly find our asymmetric loss is orthogonal to the ensemble method to some extends.

**SMSE vs REDQ (Chen et al., 2021)** We first compare our work with an ensemble Q algorithm called Randomized Ensembled Double Q-Learning (REDQ) (Chen et al., 2021). REDQ leverages an ensemble of Q-networks with additional tricks such as high update-to-data ratio and target-minimization from a subset of these Q-networks to demonstrate strong performance in a sample-efficient manner. In Tab 5 we compare the best performance (mean across 3 seeds) of REDQ and our proposed SAC-S+SMSE on some Mujoco environments and report the number of environment steps and wall-clock time duration to achieve the same. For this comparison, we use the same underlying hyperparameters such as architecture, learning rate, etc., with REDQ specifically using 10 networks, a random subset size of 2 for target construction, and an update-to-data (UTD) ratio of 20. The performance of SAC-S+SMSE is reported for the best temperature parameter selected by the procedure outlined in Appendix B.2 as done in the main paper.

REDQ indeed achieves a very high performance, which can be attributed to stable targets obtained from ensemble networks and the high UTD ratio. While a sample-efficient performance is achieved, the run-time and memory consumed by such ensemble techniques are high. If we use $N$ networks, compare $M$ networks to provide a target estimate, and perform $G$ updates per step (UTD ratio) - we can approximate the FLOPs used in REDQ for the network size used in our experiments as $\approx G \times ((2 \times \text{FwdPass} \times M) + N \times (\text{FwdPass} + (2 \times \text{BckwdPass})))$. The FLOPs count of SAC-S+SMSE can estimated with $G = 1$, $M = 1$, and $N = 1$. We obtain the FLOPS for FwdPass from fvcore library and approximate the cost of BckwdPass as $3 \times \text{FwdPass}$.

**SMSE improves Ensemble** To experiment if we can incorporate an ensemble of networks into the SMSE framework, we improve on the ensemble averaging target (AVG) baseline also used in (Chen et al., 2021), by plugging in our proposed SMSE loss to update the Q-networks from targets predisposed to overestimation bias. In Fig 11 we compare the impact of using SMSE loss instead of the baseline MSE loss. We see that the proposed loss function significantly improves the AVG baseline. To further see if our method helps reduce estimation bias we compare REDQ, AVG,

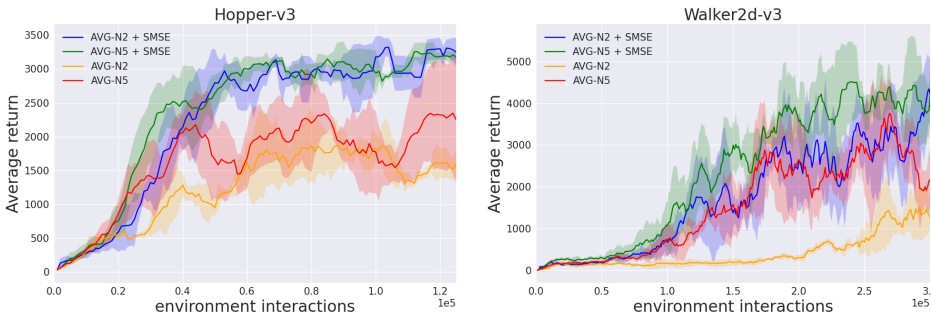

Figure 11: Impact of SMSE loss on the performance of ensemble average targets (AVG).

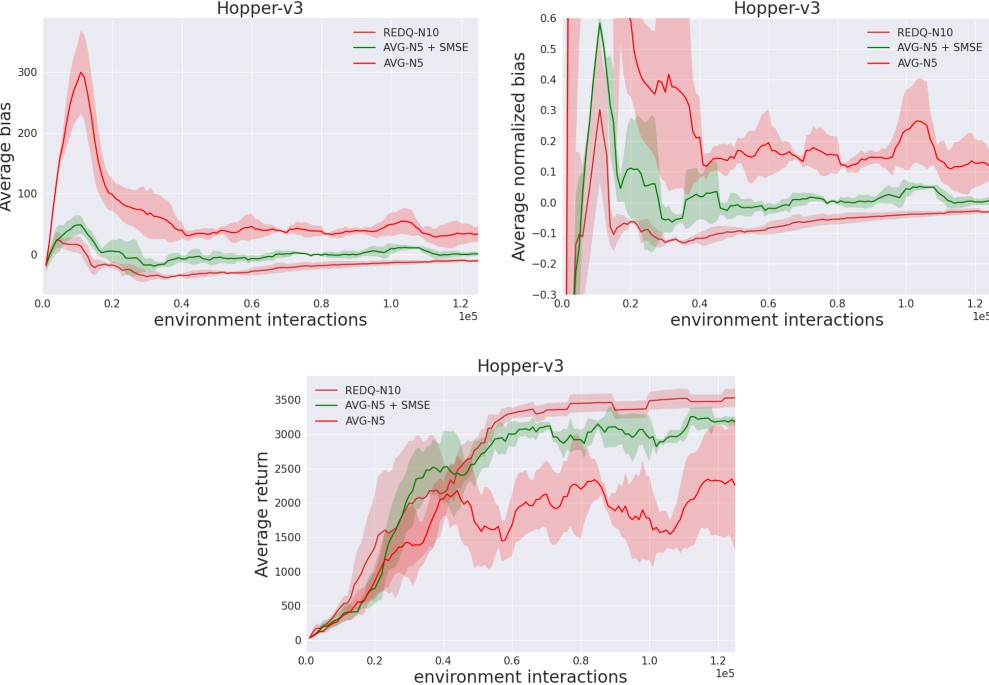

Figure 12: The plot of the top-left indicates the average value function bias (i.e. prediction - estimated true value). The plot on the top-right indicates the average normalized bias (i.e average bias divided by estimated true value). The plot at the bottom compares the performance of REDQ, AVG, and AVG + SMSE. While we observe slight underestimation in REDQ and overestimation in AVG, the average bias in AVG + SMSE is close to 0, showing that the proposed loss function effectively reduces bias in using averaged ensemble target estimates.

and AVG+SMSE on Hopper-v3 and present the results in Fig 12. Average bias is the mean of the differences between the predicted Q-values and the estimated true Q-values (estimated by a Monte Carlo simulation) over state-action pair samples. The normalized bias divides the bias by the estimated true Q-value to visualize the scale of the bias in terms of the value function. We observe that the proposed SMSE loss can effectively reduce the bias in the action-value function by keeping the average bias close to 0 and providing competitive performance improvements. These findings indicate that SMSE is better than the default MSE loss, and the improvement is orthogonal to other methods proposed to address the estimation bias problem in RL. We are optimistic that further investigation into Asymmetric loss functions can benefit many algorithms and downstream applications that rely on value functions for further optimization and planning.

