# OpenReview forum: "AsymQ: Asymmetric Q-loss to mitigate overestimation bias in off-policy reinforcement learning"
_ICLR.cc/2023/Conference — Submitted to ICLR 2023_

### Official Review · Reviewer_fW9A · 2022-10-25

**Confidence:** 4
**Correctness:** 3
**Technical Novelty And Significance:** 3
**Empirical Novelty And Significance:** 2
**Recommendation:** 5

**Clarity, Quality, Novelty And Reproducibility:**

- Clarity: Most of the paper is clear, some parts can be refined
- Quality: overall quality is good
- Novelty: the proposed modification is not super different from existing methods but can be considered somewhat novel
- Reproducibility: good, adequate details are provided


**Strength And Weaknesses:**

Strength:
- Computation efficiency: The method imposes small computation and memory overhead.
- Easy to apply: Proposed method can be applied to existing methods and does not require extensive modifications.
- Some significant results: Results on 4 Atari tasks show significant performance gain over DQN and DDQN baselines
- Technical details: many technical details are provided for better reproducibility, the computation comparison figure is great and there are studies on gradient clipping and the temparture.

Weaknesses:
- Significance of results: in Gym tasks, the proposed method can be stronger than other methods in some cases, but it is not that consistent, in some tasks, the performance gap is really small. In Atari tasks, there are only 4 tasks shown. It seems quite reliable that the proposed method can greatly improve DDPG, DQN, DDQN on given tasks. But these methods are a bit old and it is unclear how it compares to more recent methods? For example REDQ (Randomized ensembled double Q learning, ICLR 2021) on gym tasks and Rainbow variants on Atari.
- Would like to see how your method compares to SOTA algorithms on gym, for example if you follow what's in the REDQ paper, use a higher update-to-data ratio, maybe your method will have a much better performance compared to competitors? If you can reach SOTA performance with much lower computation, then that will make the performance claim much stronger.
- I believe you are tuning for each task. In that case why is it that DDPG+SMSE cannot beat DDPG in some tasks? Not sure if that really makes sense? The other thing is for the methods you are comparing to, are they also fine-tuned to each task individually?
- Figure 1 the figure is OK but you might want simply make your discussions a bit more explicit (maybe just explicitly point out early in the paper that the proposed method can be seen as providing weighted update and will make Q values faster to go down and slower to go up and be explicit about the intuition you are trying to convey in the figure)
- For figure 2, the plot is nice but the discussion on how you set up the toy example experiment is a bit vague, maybe you can rewrite some of the explanation to improve clarity. Is each curve in the figure showing what happens after many updates? And would be good to add a summary sentence in the end, just to make your insights more explicit for your reader.


**Summary Of The Paper:**

The authors propose a new method to better mitigate estimation bias. Essentially the method will allow weighted gradient updates when training the critics, and (depending on the temperature hyperparameter) give a higher weight to gradients that will lower Q value estimates and a lower weight to gradients that will increase Q value estimates. In this way a more fine-grained bias control can be achieved. A major benefit of this method is it lowers computation and memory consumption. Empirical results on Gym and some Atari tasks show it can outperform other baselines. Toy example and other empirical analysis and some theoretical support are also provided.

**Summary Of The Review:**

The proposed method is interesting, and a lot of technical details are provided which is great. My main concern is on the significance of the performance of the proposed method. All the methods compared are quite old, so it is unclear how the proposed method will perform against some of the current SOTA methods in bias reduction.

---

> ### Author Response · Authors · 2022-11-18
> **Response to review fW9A**
>
> Thank you for the detailed reviews and thoughtful feedback.
>
> **Q: Compared with REDQ**
>
> A: Based on your elaborative feedback, we conducted experiments comparing how our algorithm would fare against/with one recently proposed ensemble method, Random Ensembled Double Q Learning (REDQ). REDQ did demonstrate strong performance in a sample-efficient manner. However, we notice that is very time-consuming to train REDQ even on just a few 100K steps in contrast with non-ensemble methods. While sample efficiency is critical in real-world deployment, developing techniques to operate under computation constraints is important for edge deployment.
>
> As your insightful comment mentioned, we experimented with increasing the update-to-data (UTD) ratio for SAC-S+SMSE, but we noticed that while we see some performance improvements, it is not even comparable to the performance REDQ can achieve, which aligns with the finding in REDQ paper that reports no improvements with just increasing UTD-ratio for SAC-D baseline.
>
> To prove that our proposed technique is **orthogonal** to developments with ensemble targets. We test if our loss function can be used along with a simple average ensemble target update (AVG), a baseline used in the REDQ paper. We notice that our proposed method demonstrates significant performance improvements while reducing the estimation bias, whereas REDQ demonstrates signs of underestimation due to the use of minimization-based target construction. The performance achieved by AVG+SMSE is closer to REDQ’s performance, with fewer networks in the ensemble on the Hopper-v3 task. We report these findings in Appendix B.8, and highlight some of the findings below.
>
> - REDQ demonstrates sample efficient performance but is computationally expensive (Appendix, Table 5). We summarize the performance for Hopper-v3 environment below.
>
> | Env: Hopper-v3 | Steps | Performance | Wall-clock time (hrs) |
> | --- | --- | --- | --- |
> | REDQ | 125K | 3528.57 ± 134.14 | 20.46 |
> | SAC-S + SMSE | 1000K | 3213.84 ± 205.56 | 11.93 |
> - Using SMSE loss on average ensembled (AVG) targets can make updates that result in lower estimation bias, while REDQ updates demonstrate a slight underestimation bias for experiments on Hopper-v3 (Appendix, Figure 12)
>
> | Env: Hopper-v3 | Steps | Performance | Final Average Bias |
> | --- | --- | --- | --- |
> | REDQ (N=10) | 125K | 3528.57 ± 134.14 | -10.15 ± 2.31 |
> | AVG (N=5) + SMSE | 125K | 3180.56 ± 71.79 |   0.82 ± 2.71 |
>
> **Q: All the methods compared are quite old, so it is unclear how the proposed method will perform against some of the current SOTA methods in bias reduction.**
>
> A: In fact, most SOTA work relies on TD3 or even an ensemble of critics. Instead, our work suggests a different approach and explores whether we can achieve good performance **without multiple critics or extra computational burdens**.  To our best knowledge, no existing works report achieving good performance in this direction without ensembles. However, to address this concern, we performed additional experiments comparing it with REDQ and simple average ensemble target baselines, as reported earlier in this response.
>
> **Q: I believe you are tuning for each task. In that case why is it that DDPG+SMSE cannot beat DDPG in some tasks? Not sure if that really makes sense?**
>
> A: DDPG is indeed subsumed by the DDPG + SMSE when $T = \infty$. Since we report the best performance over a range of pre-defined temperatures that aren't very high, it does not produce a performance close to DDPG on the Swimmer-v3 and HalfCheetah-v3 tasks. It is to be noted that most prior works like TD3 have omitted results on Swimmer-v3, where DDPG provides strong performance. We include these tasks to provide a comprehensive evaluation and show that DDPG + SMSE can get close to DDPG's performance on these tasks.
>
> **Q: I believe you are tuning for each task. The other thing is for the methods you are comparing to, are they also fine-tuned to each task individually?**
>
> A: The hyperparameters for the baseline are obtained from CleanRL an open-source repository on which we base our implementation, which has extensively tuned these algorithms to work for benchmark and achieve scores comparable to the ones reported in the paper. We, therefore, rely on these parameters and present the analysis by tuning only the newly introduced hyperparameters. However, we want to bring to your attention that our proposed auto-tuning approach can be viewed as a non-fine-tuned version similar to the baseline methods, which use one set of parameters to tackle all the problems in the OpenAI gym Mujoco suite to achieve competitive performance.
>
> **Q: Improve writing and explanation for figure 1/2.**
>
> A: Thanks for the suggestions. We added more explicit discussion and highlight our intuitions.

---

### Official Review · Reviewer_h4Ht · 2022-10-30

**Confidence:** 4
**Correctness:** 2
**Technical Novelty And Significance:** 2
**Empirical Novelty And Significance:** 1
**Recommendation:** 5

**Clarity, Quality, Novelty And Reproducibility:**

The paper is clear, and the quality of writing is OK (though it can definitely be improved). The authors address reproducibility at the end of the paper.

**Strength And Weaknesses:**

Strengths: the method does appear effective, in that it does reduce overestimation bias.

Weakness:

- it is unclear if the method really improves performance. In particular, the SAC performance with two critics is lower than typical SAC performance, sot he MuJoCo resutls are not trustworthy. For DQN results, DQN is now a 10 real old algorithm. Can you show that this helps with Rainbow?

- It is not clear if fixing over-estimation should lead to better performance. Also, what about using other methods such as truncated distributional critics? How do they perform in comparison?

- No clear significance: to be honest, I am not sure what the point of the result DDPG + SMSE == SAC is. In particular, SAC is so well adopted, that unless we improve over existing state of the art algorithms, or bring new insight to the table, it is not obvious why this result should be significant in the long run.

- More experimental settings: it would be worth studying this in settings where over-estimation is likely to hurt more, like in offline RL, or in settings where we must take more gradient steps per environment step.


**Summary Of The Paper:**

This paper proposes to use asymmetric Bellman losses for TD-learning in algorithms such as DDPG, SAC, etc. The goal is to resolve overestimation bias, and the idea of the paper is intuitive. They further propose an auto-tuning algorithm for the temperature hyperparameter in their algorithm.

**Summary Of The Review:**

In summary, I think the empirical evaluation of the paper needs a lot more work. The paper needs to convince the reader why their method is significant, and why we should take note of it as a community, and this is where the paper falls short. I am opting for a reject score due to these reasons. Besides, it would also be good to see an analysis on why overestimation matters.

___

## After Rebuttal

Thanks for the responses! I am still not convinced about the significance in the long run, but I will move up my score to a 5, in light of the rebuttal.

---

> ### Author Response · Authors · 2022-11-18
> **Response to review h4Ht (part 1)**
>
> Thank you for the detailed reviews and thoughtful feedback. We want to point out that our **main contribution** is a **lightweight** method to reduce estimation bias **without increasing the computation cost**. The existing methods either demand an ensemble of critics and require more computation resources or suffer from bad performance with a single critic.
>
> **Q: The SAC performance with two critics is lower than the typical SAC performance, so the MuJoCo results are not trustworthy.**
>
> A: We respectfully disagree with the reviewer regarding the SAC performance.
>
> 1. Compared with the official codebase and [performance reported](https://github.com/rail-berkeley/softlearning/pull/127#issuecomment-602748849) from authors and codebase maintainer. Our reproduced SAC-D is better than the performance obtained from this repository. The following table report performance at  1M steps.
> |  | Official SAC Codebase | Our SAC-D |
> | --- | --- | --- |
> | Ant-v3 | 3.2k | 3.5k |
> | Walker2d-v3 | 4.2k | 4.1k |
> | Humanoid-v3 | 4.4k | 4.6k |
> | Hopper-v3 | 3.2k | 3.23k |
> 2. We base our implementations on the popular [clean RL](https://github.com/vwxyzjn/cleanrl) codebase to guarantee a fair comparison and reproducibility. Our SAC performance matches CleanRL benchmarks.
>
> **Q: It is unclear if the method improves performance. DQN is now a 10 real old algorithm**
>
> A: We show there is a lightweight method to reduce estimation bias and improve performance instead of relying on multiple Q networks, distribution critics, or other methods that incur more computations. To our best knowledge, no existing works achieve competitive performance without relying on multiple critics.
>
> We are limited by inefficient computational resources for suggested experiments and conduct the most crucial experiments that we believe can validate our novel approach.  We believe resource limitations should not stop us from making progress and exploring new directions to improve the efficiency of RL.
>
> **Q: It is not clear if fixing over-estimation should lead to better performance.**
>
> A: The main contribution of the seminal work TD3 shows that overestimation hampers reinforcement learning empirically and theoretically under proper assumptions. The proposed usage of two critics and taking a minimum of two critics is arguably the most popular approach to combat overestimation and inspired more efficient algorithms using ensemble critics. Unlike previous methods that require ensembles of critics to construct robust targets, our approach provides novel insight into how estimation bias can be corrected by altering the loss landscape without much extra computation.
>
> **Q: I am not sure what the point of the result DDPG + SMSE == SAC is.**
>
> A: We would like to highlight that the main contribution is a **lightweight method** that can reduce estimation bias **without increasing the computation cost**. SAC is a popularly adopted algorithm that utilizes the techniques developed in TD3 to construct targets considering the minimum of two critic values, which is known to suffer from underestimation bias. Accurate estimation bias can be critical for many downstream applications like planning with value functions, safe RL, affordance estimation, etc., which are particularly useful in robot learning, as mentioned in the introduction.
>
> We propose a lightweight algorithm that demonstrates that we can achieve competitive performance with a simple DDPG algorithm without needing a stochastic policy + double critic evaluation and optimization, making it an interesting candidate for edge deployment. Apart from practical applications, we believe the idea provides an interesting perspective to tackle overestimation bias, a well-known problem that the DeepRL community has been actively working to address.

---

> > ### Author Response · Authors · 2022-11-18
> > **Response to review h4Ht (part 2)**
> >
> > ****Q: More experimental settings.****
> >
> > A: Based on suggestions from the reviewer on experiments in scenarios requiring more gradient steps, we present an evaluation of how our approach can be coupled with recently proposed ensemble techniques such as Randomized Ensemble Double Q-Learning (REDQ) that demonstrates sample-efficient performance with a high update-to-data (UTD) ratio. We find that our proposed approach with a simple ensemble averaged target achieves competitive sample efficient performance and achieves very low average estimation bias. We report these findings in Appendix B.8, and highlight some of the findings below.
> >
> > - REDQ demonstrates sample efficient performance but is computationally expensive (Appendix, Table 5). We summarize the performance for Hopper-v3 environment below.
> >
> > | Env: Hopper-v3 | Steps | Performance | Wall-clock time (hrs) |
> > | --- | --- | --- | --- |
> > | REDQ | 125K | 3528.57 ± 134.14 | 20.46 |
> > | SAC-S + SMSE | 1000K | 3213.84 ± 205.56 | 11.93 |
> > - Using SMSE loss on average ensembled (AVG) targets can make updates that result in lower estimation bias, while REDQ updates demonstrate a slight underestimation bias for experiments on Hopper-v3 (Appendix, Figure 12)
> >
> > | Env: Hopper-v3 | Steps | Performance | Final Average Bias |
> > | --- | --- | --- | --- |
> > | REDQ (N=10) | 125K | 3528.57 ± 134.14 | -10.15 ± 2.31 |
> > | AVG (N=5) + SMSE | 125K | 3180.56 ± 71.79 |   0.82 ± 2.71 |

---

### Official Review · Reviewer_eb1X · 2022-10-30

**Confidence:** 4
**Correctness:** 3
**Technical Novelty And Significance:** 4
**Empirical Novelty And Significance:** 4
**Recommendation:** 8

**Clarity, Quality, Novelty And Reproducibility:**

The paper is well-written and easy to follow. It provides enough details to reproduce the results.

**Strength And Weaknesses:**

# Strength
1. The idea is exciting and novel. I haven't seen prior work that mitigates the overestimation issue via an asymmetric loss instead of an ensemble.
2. The experimental evaluation is adequate and demonstrates the advantage of the method.

# Weaknesses
1. TD3 natural decouples dynamics and model uncertainties. However, this method might also penalize good trajectories in the case of stochastic dynamics.
2. Prior work [1] has demonstrated that overestimation bias might be domain-specific. While Clipped Double Q-learning introduced in TD3 helps in some instances, it also negatively affects results. The paper can be improved by extending the experimental evaluation to these tasks.


[1] Tactical Optimism and Pessimism for Deep Reinforcement Learning
Ted Moskovitz, Jack Parker-Holder, Aldo Pacchiano, Michael Arbel, Michael I. Jordan

**Summary Of The Paper:**

The paper introduces an asymmetric loss called Softmax MSE to address the overestimation issue in value-based off-policy learning. Instead of taking a minimum of an ensemble of Q-functions, as was proposed in TD3, the authors propose penalizing overestimation in TD learning.

**Summary Of The Review:**

Overall, this is an interesting paper, and I recommend it for acceptance. There are some minor flaws that I hope the authors will address in the next revision; however, these flaws do not affect my assessment of the submission.

---

> ### Author Response · Authors · 2022-11-18
> **Response to review eb1X**
>
> Thank you for the detailed reviews and thoughtful feedback.
>
> **Q: The method might also penalize good trajectories in the case of stochastic dynamics.**
>
> A: We agree. Based on our current setup, it is hard to distinguish true-positive and false-positive improvement without more computations when we have a positive TD error. Therefore, there are some chances we put penalties on good samples.
>
> **Q: Overestimation bias might be domain-specific**
>
> A: We agree this is an interesting research topic and requires more investigation. This is not the focus of this paper; our work focuses on several popular RL tasks with a lightweight method to reduce estimation bias in off-policy learning.

---

### Official Review · Reviewer_a76y · 2022-11-03

**Confidence:** 4
**Correctness:** 3
**Technical Novelty And Significance:** 2
**Empirical Novelty And Significance:** 2
**Recommendation:** 3

**Clarity, Quality, Novelty And Reproducibility:**

The writing and presentation of the proposed method is almost clear. The proposed method is simple and effective in controlling value function bias. The proposed method is somewhat novel but similar techniques are proposed in previous related work. The advantage of the proposed method in learning performance is not significant when compared to conventional baselines, i.e., TD3/SAC.

For reproducibility, most experimental details are provided in the appendix. The source codes are also provided.


**Strength And Weaknesses:**

$\textbf{Strengths:}$
+ The writing and the presentation of the proposed method is almost clear.
+ The proposed method shows effective value function bias control without a value function ensemble.
+ The proposed method is evaluated based on multiple algorithms and in the environments with continuous and discrete actions.
+ In addition to performance comparsion, results from multiple aspects are provided, e.g., loss function landscape, value function bias comparison.

&nbsp;

$\textbf{Weaknesses:}$
- Closely related works [1-4] are not included in this paper. SUNRISE [1] also adopts an asymmetric loss function based on uncertianty quantification (which actually has an underlying connection to value function bias). [2-4] are SOTA bias control algorithms which achieve significant performance improvement over TD3 and SAC. At least, they should be included in this work and discussed.
- Although the proposed method is clear, I have a concern on using 1-step td bootstrapping target as the proxy of ground true value. I know it is non-trivial to obtain a high-quality proxy and 1-step td bootstrapping target is indeed a convenient and conventional chocie (as also used in a few prior works).
  - In my personal opinion, I do not consider that 1-step td bootstrapping target is a good proxy. As the authors also mention, “the constructed supervised target usually depends on bootstrapping as in Eq (4) and the loss is not the distance from the ground truth value function of the policy”.
  - Especially, when applying SMSE to DQN, it becomes more confusing. This is because value-based RL algorithms like DQN, follow Value Iteration rather than Policy Evaluation. In another word, 1-step td bootstrapping target is a value estimate of the improved policy rather than the current value function. I think this may also be a potential reason to the non-robust performance of DQN-SMSE as mentioned by the authors.
  - For another point, I have a little concern on the coupling of using $\delta$ for both the softmax weights and the MSE loss calculation. Since the td error is used for a proxy of under-/over-estimation bias of a specific $Q(s,a)$, I think dependent estimation may be necessary for the proxy of under-/over-estimation bias (similar to the double sampling problem).
- It is good to see the automatic temperature adjustment. However, the range of EBR needs to be predefined. The choice of the range needs to be considered in different domain. How different choices influence the learning performance is unknown. Moreover, it also introduces an additional hyperparameter $\beta_{multiplier}$.
- The proposed method DDPG-SMSE* performs infavorably to TD3/SAC-D in an overall view.

&nbsp;


Reference:

- [1] Kimin Lee, Michael Laskin, Aravind Srinivas, Pieter Abbeel. SUNRISE: A Simple Unified Framework for Ensemble Learning in Deep Reinforcement Learning. ICML 2021
- [2] Xinyue Chen, Che Wang, Zijian Zhou, Keith W. Ross. Randomized Ensembled Double Q-Learning: Learning Fast Without a Model. ICLR 2021
- [3] Arsenii Kuznetsov, Pavel Shvechikov, Alexander Grishin, Dmitry P. Vetrov. Controlling Overestimation Bias with Truncated Mixture of Continuous Distributional Quantile Critics. ICML 2020: 5556-5566
- [4] Litian Liang, Yaosheng Xu, Stephen McAleer, Dailin Hu, Alexander Ihler, Pieter Abbeel, Roy Fox. Reducing Variance in Temporal-Difference Value Estimation via Ensemble of Deep Networks. ICML 2022: 13285-13301


&nbsp;

$\textbf{Questions:}$
1) I notice that in Appendix B.5, the MC estimates of ground true value clip the trajectories at 500 steps. Can the authors justify this?
2) Also in Appendix B.5 and Algorithm 2, the text says “10,000 state-action pairs that are uniformly sampled from the replay buffer” and the buffer B is collected by the policy to evaluate. So, the buffer is an on-policy buffer for the policy evaluate, right? It also means that each policy (checkpoint) evaluates its value function bias over its own on-policy state-action supports. Do I understand it right?
3) According to Figure 3c,d and Figure 4c,d, it seems that the range of EBR [0.95, 0.98] results in a slight overestimation. Can the authors provide more explantion on this point?
4) How many seeds are used for the results in Figure 4? It seems that the stds are large since only 0.25 std is shown.


**Summary Of The Paper:**

This paper proposes a new class of policy evaluation algroithms, called AsymQ, for a lightweight and effective method to control over-/under-estimation. Different from the previous algorithms that rely on value function ensemble, the key idea of AsymQ is to adopt softmax MSE (SMSE) rather than conventional symmetric loss functions based on a proxy of over-/under-estimation bias. This transforms the landscape to emphasize the policy evaluation where overestimation occurs. The automatic adjustment of softmax temperature is designed based on a pre-defined range of effective batch ratio (EBR). The proposed method is evaluated in MuJoCo continuous control tasks, showing the effect of AsymQ in controlling value function bias under different temperature, comparable performance of DDPG-SMSE to TD3/SAC and the time cost of AsymQ. The proposed method is also evaluated based in Atari and MinAtar, demonstrating the effectiveness when applied to DQN.

**Summary Of The Review:**

According to my detailed review above, I think this paper is clearly below the acceptance threshold.

---

> ### Author Response · Authors · 2022-11-18
> **Response to review a76y (part 1)**
>
> Thank you for the detailed reviews and thoughtful feedback. We have updated our manuscript. We would like to encourage the reviewer to take a look at our revision.
>
> **Q: Closely related works[1-4] are not included in this paper.**
>
> A: We have added them to our related works in the category of ensemble methods. We want to highlight that our method takes **a different principle** than related work. Our method is lightweight as the proposed loss function change does not add much extra computation. However, we agree that comparing and contrasting our method with ensemble methods is warranted. Incorporating your and reviewer *fW9A*’s detailed feedback, we perform experiments based on a recently proposed ensemble approach, Randomized Ensemble Double Q-learning (REDQ), and present how our method compares to this approach and present how our proposed approach can be used alongside ensemble methods to obtain updates that result in lower estimation bias. We report the findings in Appendix B.8, and highlight some of the findings below.
>
> - REDQ demonstrates sample efficient performance but is computationally expensive (Appendix, Table 5). We summarize the performance for Hopper-v3 environment below.
>
> | Env: Hopper-v3 | Steps | Performance | Wall-clock time (hrs) |
> | --- | --- | --- | --- |
> | REDQ | 125K | 3528.57 ± 134.14 | 20.46 |
> | SAC-S + SMSE | 1000K | 3213.84 ± 205.56 | 11.93 |
>
> - Using SMSE loss on average ensembled (AVG) targets can make updates that result in lower estimation bias, while REDQ updates demonstrate a slight underestimation bias for experiments on Hopper-v3 (Appendix, Figure 12)
>
> | Env: Hopper-v3 | Steps | Performance | Final Average Bias |
> | --- | --- | --- | --- |
> | REDQ (N=10) | 125K | 3528.57 ± 134.14 | -10.15 ± 2.31 |
> | AVG (N=5) + SMSE | 125K | 3180.56 ± 71.79 |   0.82 ± 2.71 |
>
> **Q: 1-step TD bootstrapping target is not a good proxy for bias**
>
> A:  Robust target construction with 1-step TD error has been a well-studied direction, with many prior works suggesting using minimum over ensemble estimates, but we instead propose to introduce some inductive bias into the learning process by suggesting that the targets constructed have an inherent overestimation bias so the gradients should be weighed lower and empirically find this to be effective. However, we could consider n-step TD errors as a better proxy for estimation bias, but this comes at the additional cost of storing n-step experiences.
>
> **Q: 1-step TD bootstrapping target is a value estimate of the improved policy rather than the current value function in discrete environments**
>
> A: We agree with the reviewer's statement that the 1-step bootstrapping is an improved value estimate in the case of DQN. However, our method aims to introduce an inductive bias to counteract inflated targets due to overestimation bias. This intuition holds for both the continuous and discrete settings for learning with function approximators. So, we think it is reasonable to apply SMSE to DQN. We suspect other factors of the DQN algorithm/baseline implementation could result in a not-so-robust performance, which warrants more investigation in future works.
>
> **Q: For another point, I have a little concern about the coupling of using $\delta$ for both the softmax weights and the MSE loss calculation**
>
> A: We want to highlight that the weights for the proposed SMSE loss calculation are computed with stop-gradient and therefore do not impact the loss function with complex coupling. Intuitively it qualifies the gradient that pushes the Q-values to go up as they have an inherent overestimation bias from the constructed target.
>
> **Q: The choice of the EBR range needs to be considered in different domains. Moreover, it also introduces an additional hyperparameter** $\beta_{multiplier}$
>
> A: Similar to auto-tuning in SAC, which relies on a target entropy instead of a fixed parameter to obtain robust performance, we think that defining quantities such as EBR is considerably easier than selecting temperature annealing functions without any knowledge of its effects. So we strongly believe it is easier to specify such ranges with an auto-tuning algorithm rather than tune the temperature parameter to control the loss landscape. We additionally provide results (Appendix, Figure 9) indicating that the sensitivity of  $\beta_{multiplier}$ on the performance is low.
>
> **Q: The proposed method, DDPG-SMSE\*, is unfavorable to TD3/SAC-D in an overall view.**
>
> A: We respectfully disagree with this statement, DDPG-SMSE* produces performance on par with TD3/SAC-D on challenging tasks like Ant-v3 [DDPG-SMSE*: 5.42, TD3: 5:37, SAC-D: 3.83] (x10^3) and Humanoid-v3 [DDPG-SMSE*: 5.15, TD3: 5.34, SAC-D: 5.00] (x10^3). It is to be noted that most prior works have omitted to present results on Swimmer-v3 and HalfCheetah-v3, where DDPG provides strong performance; we have included these tasks for a comprehensive evaluation.

---

> > ### Author Response · Authors · 2022-11-18
> > **Response to review a76y (part 2)**
> >
> > **Q: MC estimate of the ground truth value clip the trajectories at 500 steps. Can the authors justify this?**
> >
> > A: Empirically, we can find accumulated discounted rewards after the 500th step is very small because $\gamma^{n}$ is very small when $n$ is greater than 500. Prior works like REDQ have used around **350** steps to cut off the MC sampling to estimate the ground truth value function.
> >
> > **Q: So, the buffer is an on-policy buffer for the policy evaluation right? It also means that each policy (checkpoint) evaluates its value function bias over its on-policy state-action supports.**
> >
> > A: Yes, the buffer is collected by performing an epsilon-greedy exploration with the checkpoint policy. So the state-action pairs over which the bias is evaluated are based on the states explored on policy with noise.
> >
> > **Q: it seems that the range of EBR [0.95, 0.98] results in a slight overestimation. Could the authors explain this point more?**
> >
> > A: Yes, with a temperature keeping EBR in the range [0.95, 0.98], we see some slight overestimation in Walker2d (Figure 3). While further temperate ranges could have been explored to reduce the bias further, we chose to settle with this range to use in the auto-tune algorithm as it did produce effective performance (average evaluation reward) across all tasks in the Mujoco benchmark when compared with popular baselines like TD3 and SAC and report the findings as is without more hyperparameter searches.
> >
> > **Q: How many seeds are used for the results in Figure 4? It seems that the stds are large since only 0.25 std is shown.**
> >
> > A: We believe the reviewer is referring to Figure 3. The evaluation is performed over 5 seeds, and the standard deviation scale of 0.25 is chosen only for sub-figures 3c and 3d to better highlight the trends in the average estimation bias. Sub-figures 3a and 3b reflect 1-std of the estimation bias evaluated on 10,000 states.

---

### Decision · Program_Chairs · 2023-01-20

**Decision:**

Reject

**Justification For Why Not Higher Score:**

Insufficient experimental results to support claims

**Justification For Why Not Lower Score:**

N/A

**Metareview: Summary, Strengths And Weaknesses:**

This paper proposes the use of asymmetric losses to reduce the overestimation bias in off-policy reinforcement learning. The proposed approach is novel and elegant, but weaknesses in the supporting experiments led to reviewers questioning the effect size of the approach particularly in comparison to more recent baseline algorithms. Whilst the current experiments were not fully convincing to the majority of reviewers, there was broad acknowledgement of the novelty of the approach and early signs that this could be a promising direction for future work.